# Development of a parametrised atmospheric $NO_x$ chemistry scheme to help quantify fossil fuel $CO_2$ emission estimates

Chlöe N Schooling<sup>1</sup>, Paul I Palmer<sup>1,2</sup>, Auke Visser<sup>3</sup>, and Nicolas Bousserez<sup>3</sup>

Corresponding author: Chlöe N Schooling (cschooli@ed.ac.uk) and Paul I Palmer (pip@ed.ac.uk)

## Abstract.

Success of the Paris Agreement relies on rapid reductions in fossil fuel CO<sub>2</sub> (ffCO<sub>2</sub>) emissions. Atmospheric data can verify the ffCO<sub>2</sub> reductions pledged by nations in their nationally determined contributions. However, estimating ffCO<sub>2</sub> from atmospheric CO2 is challenging due to natural fluxes and varying backgrounds. One approach is to combine with nitrogen oxides (NO<sub>x</sub> = NO + NO<sub>2</sub>), which are co-emitted with CO<sub>2</sub> during combustion. A key challenge in using NO<sub>x</sub> to estimate ffCO2 is the computational cost of modelling atmospheric photochemistry. Additionally, the NO2:NO column ratio must be well understood to convert model NO<sub>x</sub> columns to NO<sub>2</sub> columns for comparison with satellite data. We use random forest regression to parameterise NO<sub>x</sub> chemistry, relying only on meteorological parameters and NO<sub>x</sub> concentration. The regression is trained on outputs from a nested GEOS (Goddard Earth Observing System)-Chem model simulation for mainland Europe in 2019. We develop a monthly NO<sub>x</sub> chemistry parameterisation that performs well when tested on perturbed emission runs  $(R^2 > 0.95)$  and on unseen meteorology for 2021  $(R^2 > 0.79)$ . We also parameterise the NO<sub>2</sub>:NO ratio  $(R^2 > 0.99)$  on perturbed outputs,  $R^2 > 0.92$  on unseen meteorology). Additionally, we present an alternative method to predict  $NO_x$  rates by scaling baseline  $NO_x$  rates with changes in  $NO_x$  concentration ( $R^2 = 1.0$  on perturbed outputs). Our models reproduce  $NO_2$  columns with minimal deviation from full-chemistry models, with reconstruction error smaller than the TROPOspheric Monitoring Instrument (TROPOMI) precision in over 99.9% of cases, supporting robust ffCO<sub>2</sub> inversion efforts. These results provide a robust framework for accurately estimating fossil fuel CO2 emissions from atmospheric data, enabling more reliable monitoring and verification of global emissions reductions.

# 20 1 Introduction

Reaching net zero greenhouse gas emissions is a global goal, needed to curb further warming of our planet. Achieving that goal on a national scale requires accurate knowledge about fossil fuel emissions of  $CO_2$  (ff $CO_2$ ) to verify a country's progress towards achieving their Nationally Determined Contributions under the Paris Agreement. But how can a country assess whether they are heading in the right direction? The default approach is to use national inventories that are compiled from energy

<sup>&</sup>lt;sup>1</sup>School of GeoSciences, University of Edinburgh, United Kingdom

<sup>&</sup>lt;sup>2</sup>National Centre for Earth Observation, Leicester, United Kingdom

<sup>&</sup>lt;sup>3</sup>European Centre for Medium-Range Weather Forecasts, Bonn, Germany

statistics and emission factors but they are uncertain for various reasons, mainly associated with the veracity of the statistics and their spatial and temporal distributions and the default assumption of time-invariant emission factors (Kuenen et al., 2014; Hoesly et al., 2018). Such 'bottom-up' inventories are typically available with a delay of 2 years (Janssens-Maenhout et al., 2019) thereby introducing a temporal disconnect between climate action and results. The alternative 'top-down', data-driven approach uses Bayes' theory to infer  $CO_2$  emission estimates from observed changes in atmospheric  $CO_2$ . This approach is also subject to uncertainties including errors in atmospheric transport models, sparse observational coverage, and background concentration estimation (Peylin et al., 2013; Andrew, 2020). One of the remaining challenges associated with this atmospheric approach is isolating the combustion and natural contributions to atmospheric  $CO_2$  (Oda et al., 2023). Various approaches have been proffered to address that challenge, which fall into two broad categories: spatial disaggregation of combustion (Shu and Lam, 2011; Liu et al., 2018) and natural fluxes and using an additional trace gas (Meijer et al., 1996; Lopez et al., 2013; Wenger et al., 2019; Super et al., 2020), associated exclusively with combustion or natural processes common to  $CO_2$ . One such trace gas is  $NO_x$ , but due to the large computational overhead of directly modelling the atmospheric  $NO_x$  photochemistry, we endeavor to determine an alternative methodology to model  $NO_x$  chemistry. Here we describe a parameterisation of tropospheric nitrogen oxide ( $NO_x = NO + NO_2$ ) chemistry that effectively unlocks our ability to use1  $NO_x$  alongside  $CO_2$  to quantify ff $CO_2$  estimates within an Bayesian inference framework, particularly in the context of an operational system.

30

40

Extracting energy from carbon-based fuels relies on breaking apart atomic bonds that form the molecular structure of the fuel, thereby releasing energy. This is achieved by combustion in which the fuel, composed primarily of hydrogen-carbon bonds, is oxidized by molecular oxygen (O<sub>2</sub>). Generally, more energy is released during combustion for fuels with a higher H:C ratio. The primary products of combustion are CO<sub>2</sub> and water vapour. However, when combustion is inefficient – for example, due to insufficient  $O_2$  to fully oxidise the fuel – a wider range of compounds is released, depending on the composition of the fuel being burned. For many combustion processes, air is used to provide O2. While molecular nitrogen (N2) in air does not take part in the combustion reaction, the high temperatures involved can thermally dissociate N<sub>2</sub> to facilitate the production of NO (and to a lesser extent  $NO_2$ ), which is subsequently co-emitted with the  $CO_2$  emissions. The advantage of using atmospheric  $NO_x$ as a tracer of ffCO<sub>2</sub> is its relatively short lifetime, on the order of hours to days, which means that we can link elevated NO<sub>2</sub> satellite columns directly to their parent NO<sub>x</sub> emissions. Numerous studies are using observations of NO<sub>x</sub> and NO<sub>2</sub> to constrain estimates of ffCO<sub>2</sub> (Berezin et al., 2013; Lopez et al., 2013; Goldberg et al., 2019; Super et al., 2020). With the increasing availability of in situ and satellite measurements of atmospheric CO<sub>2</sub>, NO<sub>2</sub> and other fossil-fuel tracers, deriving ffCO<sub>2</sub> through multi-species model inversion techniques is becoming a widely used approach (Feng et al., 2009; Nayagam et al., 2023; Super et al., 2024; Wang et al., 2025). However, a key limitation of this method is the uncertainty in CO<sub>2</sub>:NO<sub>x</sub> emission ratios, which vary by sector, fuel type, and combustion technology (Jiang et al., 2010; Wang et al., 2025). Additional challenges include errors in atmospheric transport modelling, accurate representation of chemical processes, and limited observational coverage.

We present a methodology for parameterising  $NO_x$  chemistry to reduce the associated computational overhead. We consider  $NO_x$  because its constituents, NO and  $NO_2$ , rapidly interconvert (Jacob, 1999). By modelling  $NO_x$  as a proxy for the combined NO and  $NO_2$  we can save a considerable amount of computational time that would otherwise be spent on photochemical calculations (previously shown in Wu et al. (2023)). To do this we need a model that can predict the net loss of  $NO_x$  at each

time step and grid point. The rate of decay of NO<sub>x</sub> is driven by a number of meteorological parameters (Nguyen et al., 2022) including, but not limited to, the irradiance from sunlight, air temperature and solar zenith angle. In this study, we develop a machine learning-based random forest regression model, trained on a full-chemistry version of the GEOS (Goddard Earth Observing System)-Chem atmospheric chemistry model, to accurately predict the atmospheric NO<sub>x</sub> rate of change using a small set of driving variables. We evaluate the robustness of our parameterised NO<sub>x</sub> chemistry using perturbed emissions on the order of those we typically employ in ensemble Kalman filter techniques. With atmospheric inversion methods in mind, atmospheric NO<sub>x</sub> emission estimates tend to be constrained by satellite column observations of NO<sub>2</sub> (Napelenok et al., 2008; Zhao and Wang, 2009; Kemball-Cook et al., 2015) so our parameterised model must also be able to describe changes in NO<sub>2</sub>. We achieve this by developing a further random forest-based model, which can predict the species concentration NO<sub>2</sub>:NO ratio.

Figure 1 shows a schematic overview of the steps used to parameterise NO<sub>x</sub> chemistry and partitioning for efficient modelling of NO<sub>2</sub> columns. The first stage involves running atmospheric simulations of NO<sub>x</sub> using offline chemistry rates, which are either predicted by random forest models (described in section 2.2) or estimated through relative scaling (described in section 2.3). In the second stage, the NO<sub>x</sub> output from these simulations is converted to NO<sub>2</sub>, enabling direct comparison with satellite observations such as TROPOMI NO<sub>2</sub>. This approach provides an efficient framework suitable for data assimilation applications.

In the next section, we describe the GEOS-Chem atmospheric chemistry transport model that we use to train our random forest models, the satellite observations of column  $NO_2$  that we use to evaluate our parameterised atmospheric chemistry model for  $NO_2$ , and the approach we take to construct the random forest model. In section 3, we report the performance of random forest models of atmospheric  $NO_2$  and  $NO_2$ :NO, and evaluate the corresponding atmospheric  $NO_2$  columns using satellite data. We conclude the paper in section 4.

## 2 Data and methods

Here, we describe the GEOS-Chem atmospheric transport model used to build our random forest regression models, the satellite column data we use to evaluate our parameterised model of atmospheric  $NO_x$  chemistry, and details that describe how we develop our random forest regression models. A random forest regression model, or a constant lifetime scaling based approach can be used to predict the chemistry rates. The modelled  $NO_x$  concentrations are then converted to  $NO_2$  using an additional random forest model. This efficient approach significantly reduces GEOS-Chem's computational cost for forward modelling of  $NO_2$  columns. This is particularly useful for high resolution data assimilation, allowing anthropogenic  $NO_x$  emission perturbations to be compared with satellite  $NO_2$  observations, such as the TROPOspheric Monitoring Instrument (TROPOMI).

## 90 2.1 GEOS-Chem atmospheric chemistry transport model


We use version 14.2.2 of the GEOS-Chem atmospheric chemistry transport model (Bey et al., 2001) to describe the emissions, transport, and chemical production/loss of atmospheric  $NO_x$ . For the purpose of our study, we use a nested version of the full chemistry model, centred over mainland Europe (32.75 to 61.25° N, -15 to 40 ° E) with 47 vertical levels, approximately 30 of which fall below the dynamic tropopause, where the first model layer has a depth of 130-180 m. The nested model runs with a horizontal spatial resolution of  $0.25^{\circ}x0.3125^{\circ}$ . Initial conditions and lateral boundary conditions to the nested domain were created from a consistent global version of the GEOS-Chem model run at  $4^{\circ}\times5^{\circ}$ , with three-hourly output fields. We ran the model with a transport timestep of 5 minutes and a chemistry timestep of 10 minutes.

The model is driven by offline meteorology fields from the GEOS Forward Processing (GEOS-FP) product from the Global modelling and Assimilation Office (GMAO) at NASA Goddard Space Flight Center. GEOS-FP has a native horizontal resolution of  $0.25^{\circ} \times 0.3125^{\circ}$  with 72 vertical pressure levels and 3 hr temporal resolution. To describe the emissions of  $NO_x$  we used anthropogenic emissions from the Community Emissions Data System (CEDS) version 2 (Hoesly et al., 2018), which provides NO emissions for anthropogenic combustion (industry, energy extraction), and non-combustion sources (agriculture, solvents), including surface transport and shipping. Aircraft emissions for NO and  $NO_2$  are taken from the Aviation Emissions Inventory Code (AEIC) (Simone et al., 2013). Pyrogenic emissions of NO are taken from the Global Fire Emissions Database (GFED) version 4.1 (Randerson et al., 2017).

GEOS-Chem's full-chemistry mechanism simulates atmospheric chemistry by explicitly solving a comprehensive network of chemical reactions, capturing the production, transformation, and loss of  $NO_x$  and related species.  $NO_x$  chemical loss is

Figure 1. A schematic illustrates how  $NO_x$  chemistry parameterisation models are integrated into GEOS-Chem for modelling of atmospheric  $NO_x$  without a full chemistry scheme.

simulated through key reactions such as  $NO_2$  reacting with ozone  $(O_3)$  to form  $NO_3$ , hydroxyl radicals (OH) to produce nitric acid  $(HNO_3)$ , and hydroperoxyl radicals  $(HO_2)$  to form peroxynitric acid  $(HNO_4)$ . Organic nitrate formation is included through the reactions of  $NO_2$  with methyl peroxy radicals  $(MO_2)$  and methacryloyl peroxy radicals  $(MCO_3)$ , forming methyl peroxy nitrate (MPN) and peroxyacetyl nitrate (PAN), respectively. Additional loss occurs via  $NO_3$  reacting with  $NO_2$  to produce dinitrogen pentoxide  $(N_2O_5)$ . Simultaneously, the model accounts for important regeneration pathways, including the thermal decomposition of  $N_2O_5$  into  $NO_3$  and  $NO_2$ , the breakdown of PAN to release  $NO_2$  and methacryloyl peroxy radicals  $(MCO_3)$ , and the photolysis of  $HNO_4$  to produce  $NO_2$  and  $HO_2$ . Rapid NO to  $NO_2$  exchange is simulated through key reactions, including  $NO + O_3 \longrightarrow NO_2 + O_2$ , which relies on ozone to oxidize NO, and  $NO + NO_3 \longrightarrow 2$   $NO_2$ , which occurs through the reaction of nitric oxide with nitrate radicals. Additionally, photochemical reactions driven by sunlight include  $NO_2 + O_2 + hv \longrightarrow NO + O_3$ , where nitrogen dioxide photodissociates to form nitric oxide. The mechanism determines reaction rates using reaction rate coefficients that depend on temperature, pressure, and solar radiation, alongside environmental inputs like meteorological fields and species concentrations.






The average diurnal cycle of  $NO_x$  chemical rate of change calculated from full-chemistry simulations is presented in Fig. A1 for the four seasons of the year. The shape of the diurnal cycle in the  $NO_x$  tendency varies seasonally, influenced by changing sunlight intensity and atmospheric conditions. In winter, the net  $NO_x$  loss peaks predominantly at night, when photolytic regeneration ceases and reservoir species like  $HNO_3$  and PAN accumulate, removing  $NO_x$  from the reactive pool. During spring and autumn, while a nighttime peak loss remains, there is an additional peak of comparable magnitude in the morning around 0900–1000 local solar time (LST). In summer, the maximum net loss shifts to the early morning hours 0700–0800 LST, likely driven by rapid photochemical activity as sunlight increases. Meanwhile, by the afternoon we find episodes of net  $NO_x$  production, reflecting stronger photolytic regeneration under high solar intensity. These seasonal and diurnal variations reflect complex interactions between photochemistry, emission patterns, and atmospheric transport, resulting in shifts of  $NO_x$  sinks and sources throughout the day and year.

The  $NO_x$  concentration, the  $NO_x$  chemical rates of change, and relevant meteorology were output at a temporal resolution of one hour. The chosen meteorological parameters are shown in Table 1. These were selected as they were all found to have a relationship with the net  $NO_x$  chemical rate of change.

The model was run for the full year 2019 with baseline (unperturbed)  $NO_x$  anthropogenic emissions taken from the CEDs emission inventory. This data was used to train the regression models. To further validate the regression model's performance under varying emissions, additional model runs were conducted with random perturbations applied to anthropogenic  $NO_x$  emissions on the order of  $\pm 20\%$ . We chose this size of perturbation because a 20% increase in emissions induces changes in  $NO_2$  columns on the same order of magnitude as the difference observed between GEOS-Chem and TROPOMI (as in Fig. 2a). These perturbed runs were performed for 10 days in January, April, July, and October. A model run for the year 2021 was also performed in order to test the regression performance for an unseen meterological period.

Figure 2. a) Sensitivity testing shows that the impact of 20% emission perturbations on modelled  $NO_2$  columns is on the same order as the deviations between GEOS-Chem and TROPOMI. (b) The impact of emission perturbations on the  $NO_x$  chemistry rate becomes negligible (<1% change, or  $\Delta NO_x$  rate <  $9\times10^3$  molec/cm<sup>3</sup>/s) above 3km from the ground. Additionally, chemistry rate change is negligible in all cases where  $\Delta NO_x$  concentration <  $5\times10^4$  molecules/cm<sup>3</sup>.

| Parameter   | Description                      | Units                              | Rate predicition | Ratio predicition |
|-------------|----------------------------------|------------------------------------|------------------|-------------------|
| $NO_x$      | Species concentration            | $\mathrm{molec}\;\mathrm{cm}^{-3}$ | ✓                | ×                 |
| SZA         | Solar zenith angle at grid point | degrees                            | ✓                | $\checkmark$      |
| Longitude   | Grid point coordinate            | degrees-East                       | ✓                | $\checkmark$      |
| Latitude    | Grid point coordinate            | degrees-North                      | ✓                | $\checkmark$      |
| Altitude    | Height above ground level        | m                                  | ✓                | ✓                 |
| Radiation   | Incident short wave radiation    | ${ m W}{ m m}^{-2}$                | ✓                | ✓                 |
| Temperature | Atmospheric temperature          | K                                  | ✓                | ✓                 |
| Humidity    | Water vapour mixing ratio        | $vol\ vol^{-1}$                    | ✓                | ✓                 |
| Wind speed  | Wind speed magnitude             | ${ m m~s^{-1}}$                    | ✓                | ✓                 |
| Density     | Dry air density                  | ${\rm kg}~{\rm m}^{-3}$            | ×                | ×                 |
| PBL height  | Planetary boundary layer height  | m                                  | ×                | ×                 |
| Pressure    | Air pressure                     | hPa                                | ×                | ×                 |
| CO          | Carbon monoxide dry mixing ratio | $vol\ vol^{-1}$                    | ×                | ×                 |
| $O_3$       | Ozone dry mixing ratio           | $vol\ vol^{-1}$                    | ×                | ×                 |

**Table 1.** Input parameters selected through forward feature selection for random forest regression models predicting the  $NO_x$  chemical net rate of change [molec cm<sup>-3</sup> s<sup>-1</sup>] and the  $NO_2$ :NO partitioning ratio.

# 140 2.2 Random Forest regression modelling




We trained two random forest regressor models to predict the  $NO_x$  net chemical rate of change, and the  $NO_2$ :NO partitioning ratio. Random forest models are an ensemble machine learning method, which combine the predictions of many decision trees to improve accuracy and reduce overfitting (Breiman, 2001). A decision tree is a simple predictive model that makes a series of splits in the data based on input variables. At each node, the algorithm chooses the predictor and threshold that best separate the data with respect to the target, continuing until each final branch (or "leaf") gives a prediction. While a single tree is easy to interpret, it can overfit the data. Random forests address this by building a "forest" of many trees, each trained on a random subset of the data and predictors. This randomness ensures the trees capture diverse patterns, and averaging their outputs yields more robust predictions. Such an algorithm is well-suited to this study as, unlike traditional regression approaches, it does not require assumptions about linearity and can flexibly capture complex relationships and interactions between meteorological drivers and chemical tendencies. Additionally, random forests are relatively computationally efficient to train and can handle correlated predictor variables, making them well suited for large atmospheric datasets.

These models were built using the Sci-kit learn python package (Pedregosa et al., 2011). We evaluated model performance using the coefficient of determination ( $\mathbb{R}^2$ ), which quantifies the proportion of variance explained by the model; the mean absolute error (MAE), which measures the mean magnitude of prediction errors; and the mean bias, which indicates the mean tendency of the model to overpredict or underpredict relative to observations. These are defined by the following equations, where  $y_i$  are true values,  $\hat{y}_i$  are predicted values,  $\bar{y}$  is the mean of the true values, and N is the number of datapoints:

$$R^{2} = 1 - \frac{\sum_{i=1}^{N} (y_{i} - \hat{y}_{i})^{2}}{\sum_{i=1}^{N} (y_{i} - \bar{y})^{2}} \qquad \text{MAE} = \frac{1}{N} \sum_{i=1}^{N} |y_{i} - \hat{y}_{i}| \qquad \text{Mean Bias} = \frac{1}{N} \sum_{i=1}^{N} (y_{i} - \hat{y}_{i})$$
(1)

We separately trained both regression models for each month of the year, for which we report results from January, April, July, and October 2019. The models were developed using the NO<sub>x</sub> concentration, the spatial location and a range of meteorological variables as input parameters. We considered a total of 14 input parameters as predictors in the models, shown in table 1.

To identify the most relevant features for the models, we performed a comprehensive forward selection wrapper procedure, which iteratively adds the feature that yields the largest improvement in mean absolute error until no further gain is observed. Figs. A2a and A2b detail how the performance of the models changed as we added features for the prediction of chemistry rate and the partitioning ratio, respectively. Based on this procedure, we selected a set of nine features for the chemistry rate model, and eight features for the partitioning ratio model (presented in Table 1). Five of the parameters; air pressure, air density, height of the planetary boundary layer, and the mixing ratio of ozone (O<sub>3</sub>) and the mixing ratio of carbon monoxide (CO), were consistently excluded from all models during feature selection. The respective importance of each feature across both models for the four months studied are plotted in Fig. A2c. For the chemistry rate prediction, the NO<sub>x</sub> concentration and the solar zenith angle are consistently emerge as the most important predictorss, contributing around 70% of the total feature importance in the model. In the ratio prediction, solar zenith angle, altitude, and temperature are the primary predictors during the colder months (January and October), while temperature alone serves as the dominant predictor in the warmer months

(April and July). Additionally, the impact on model performance of removing each of the 14 parameters in turn is presented in Fig. A2d. The individual relationship between the nine selected predictors and the NO<sub>x</sub> chemistry rate of change are shown in Fig. A3.

To avoid unnecessarily complex models, we tuned the model hyperparameter values to optimise the trade-off between computational efficiency and prediction accuracy. Specifically, we conducted a grid search across the four main hyperparameters in the random forest regression model: the number of trees (estimators), maximum tree depth, maximum number of leaf nodes, and maximum number of features considered at each split. We selected each hyper-parameter as the value at which performance plateaued, defined here as the point beyond which further increases in the parameter resulted in less than a 2% improvement in model performance. The results of the such tuning are presented in Fig. A4. The final optimised model achieved a prediction time of 6 ms per sample, providing a good balance between accuracy and computational cost. In addition to reducing computational time, simplifying the random forest by limiting tree complexity and number also reduces the risk of overfitting, thereby improving the generalisability of the model to new data.

We trained and tested our  $NO_x$  chemistry regression models on model grid points in the first 3 km above the surface – the region where changes to surface emissions were found to directly influence the atmospheric chemistry, see Fig. 2b. The regression model for the  $NO_2$ :NO ratio was predicted for each level in the troposphere, and trained on the subset of model data that coincides with the TROPOMI swath (11:30 - 15:30 LST overpass). The  $NO_2$ :NO ratio can be used to convert the concentration of  $NO_x$  to  $NO_2$ :

90 
$$NO_2 = NO_x \frac{NO_2 : NO}{1 + NO_2 : NO}$$
. (2)

To evaluate model generalisability, we tested model performance using two complementary approaches. Primarily, we assessed predictions on unseen emission perturbation scenarios while holding meteorology fixed. Specifically, we focused on  $\pm 20\%$  emission perturbations similar to those used in ensemble Kalman filter applications (Feng et al., 2009, 2023). This isolates the model's responsiveness to emission changes under consistent atmospheric conditions and reflects its intended use in inversion frameworks, where emissions are perturbed while meteorology remains prescribed. In addition, we include in the appendix (Fig A6) an evaluation on an entirely independent simulation run for the year 2021, representing unseen meteorological conditions due to its different temporal period. For both approaches, training and testing datasets were constructed via random sampling across all spatial locations and time steps. The training set comprised a random 10% subset of the unperturbed data, while the test set comprised 0.25% of the perturbed (or 2021) data, ensuring minimal overlap in specific spatiotemporal conditions. Combined, this dual testing strategy rigorously evaluates the models' ability to generalise across both emission changes and meteorological variability, providing confidence in their performance for atmospheric inversion applications.

# 2.3 NO<sub>x</sub> chemical lifetime





In an alternative formulation, we apply the assumption that the effective lifetime of atmospheric  $NO_x$  remains constant under stable meteorological conditions. Hence, if a full chemistry model run is available for a baseline emission scenario, the chem-

istry rates for perturbed scenarios can be calculated by scaling the original rate according to the proportional change in NO<sub>x</sub> concentration. This approach serves as an alternative to using regression models for predicting the chemistry rates.

The effective atmospheric lifetime,  $\tau$  of NO<sub>x</sub> is given by:



$$\tau = \frac{\text{NO}_{\text{x}}}{R_{\text{NO}_{\text{x}}}},\tag{3}$$

where  $NO_x$  denotes the combined NO and  $NO_2$  species concentrations [molec cm<sup>-3</sup>] and  $R_{NOx}$  is the instantaneous chemical rate of net loss [molec cm<sup>-3</sup>s<sup>-1</sup>], which accounts for the balance between its chemical production (e.g., from reactions involving NO or  $NO_2$  precursors) and its chemical loss processes (e.g., reactions forming reservoirs like  $HNO_3$  or NOy species). Note that when  $NO_x$  experiences an instantaneous net chemical production, this effective atmospheric lifetime becomes negative. We advise the reader that this effective lifetime does not represent an intrinsic first-order decay timescale for  $NO_x$ . Instead, it provides a practical framework to express net rates of change relative to the amount of  $NO_x$  present, which we find to be an intrinsically stable metric. The benefit of looking at the effective chemical lifetime, rather than the net rate of change, is that the quantity is largely independent of species concentration. This independence allows for a more stable understanding of the  $NO_x$  chemistry, irrespective of fluctuations in its concentration caused by emission changes.

We found that while the influence of  $\pm 20\%$  emission perturbations cause clear changes to the  $NO_x$  chemical net rate of change, the resulting changes to atmospheric lifetime are considerably smaller (see Fig. A5). This result suggests that the chemical lifetime is driven by the meteorology and location in the model but is less sensitive to changing concentrations of  $NO_x$ . The unperturbed model run provides  $NO_x$  concentrations and rates of change at a 1-hour temporal resolution, allowing the chemical rate of change to be updated every hour under the assumption of an unchanged chemical lifetime. The new rate of change can be determined using the  $NO_x$  lifetime,  $\tau$ , and the local  $NO_x$  concentration:

$$R_{\text{NO}_{x}}(x, y, z, t) = \frac{\text{NO}_{x}(x, y, z, t)}{\tau(x, y, z, t)}.$$
(4)

For this method, an initial unperturbed full-chemistry model run must be employed to determine the NO<sub>x</sub> chemical lifetime τ(x,y,z,t) for each grid-point and time-point for the spatial and temporal region of interest. Then for any further perturbed model runs, the chemistry rates can be determined without the need of an integrated chemistry scheme, thereby saving considerable computational time. The updated chemistry rates are then simply scaled by the ratio of the new NO<sub>x</sub> concentration to the original NO<sub>x</sub> concentration; so, if the concentration doubles then we assume a doubling in the net chemical rate of change.

This method for updating the NO<sub>x</sub> chemistry is referred to as the constant lifetime scaling-based method.

#### 2.4 Regression-based atmospheric chemistry transport modelling

For this study, we added the  $NO_x$  species to the GEOS-Chem tagged carbon model,  $CO_2$ , CO, methane, and carbonyl sulphide, in which individual tagged tracers track contributions of these trace gases from geographical regions and/or natural and human-driven fluxes. This model does not include an integrated chemistry scheme and therefore the  $NO_x$  species chemical rate of change is determined using the  $NO_x$  chemistry regression model. Going forward, we refer to this model as the regression-based atmospheric chemistry transport model (shown in Fig. 1).

We performed a full-chemistry model run with emission perturbations to evaluate the impact of emission changes on  $NO_x$  chemistry, and later to assess the performance of our regression model in predicting the effects of emission changes. An analysis of how the emission-driven changes in chemistry rate varied with the atmospheric altitude as well as the change in  $NO_x$  concentration is shown in Fig. 2b. The net rate of change in  $NO_x$  chemistry showed minimal variability at altitudes above 3 km, where the chemistry change was less than  $9\times10^3$  molec/cm<sup>3</sup>/s. Additionally, minimal variability in atmospheric chemistry was observed when the absolute change in  $NO_x$  concentration was less than  $5\times10^4$  molec/cm<sup>3</sup>, which corresponds to a chemistry change of less than  $2\times10^3$  molec/cm<sup>3</sup>/s. Based on these findings, we set a condition to update the  $NO_x$  net chemical rate of change using the unperturbed full-chemistry outputs for altitudes above 3 km and for regions where the change in  $NO_x$  concentration is less than  $5\times10^4$  molec/cm<sup>3</sup>. For all other regions, the chemistry regression model is used to predict the new rate of change.

We also used the constant lifetime scaling method (see above) to predict the new rate of change. Looking to Fig. 1 we can see that this methodology provides an alternative approach to the regression-based atmospheric chemistry model for modelling  $NO_x$  columns. Throughout this paper we will compare the results of the regression-based chemistry scheme and the constant lifetime scaling-based approach.

We ran the model for 10 days in January, April, July, and October which provided contrasting seasonal conditions to test the model. For each run, we use the  $\pm 20\%$  perturbed anthropogenic  $NO_x$  emission sets. To evaluate the veracity of the  $NO_x$  column model outputs for the regression-based chemistry model and for the constant lifetime scaling model, we compare them with the full-chemistry model outputs. We use our  $NO_2$ :NO ratio regression model to convert  $NO_x$  results from our atmospheric chemistry regression model to  $NO_2$  columns, sampled at the time and location of TROPOMI data, so they can be compared with TROPOMI  $NO_2$  column data.

# 2.5 TROPOMI satellite column observations of NO<sub>2</sub>







We use TROPOMI NO<sub>2</sub> tropospheric columns (S5P Level 2, product version 2.2.0, processing version 1.6.0.) to compare with the GEOS-Chem model output (see Fig. 1). TROPOMI was launched in 2017 in a Sun-synchronous orbit with a local equatorial overpass time of 13:30. It has a swath width of 2600 km and a ground pixel of  $7 \times 7$  km<sup>2</sup> in the nadir. Due to the width of the swath, the 13:30 overpass time corresponds to data captured with local solar time (LST) ranging from 11:30 and 15:30 in the highest latitude regions of the European domain. We only used data with a quality flag  $\geq$  0.75, filtering out data affected by elevated cloud cover, aerosol loading, and larger solar and viewing zenith angles. We analysed TROPOMI data for 10 days in January, April, July, and October 2019.

For our study, we regridded TROPOMI data to our  $0.25^{\circ} \times 0.3125^{\circ}$  GEOS-Chem model grid. To enable a comparison between TROPOMI and GEOS-Chem, we sampled the model at the location and time of each TROPOMI observation. We applied scene-dependent TROPOMI averaging kernels, describing the instrument sensitivity to changes in atmospheric  $NO_2$ , to the corresponding model  $NO_2$  profiles.

## 3 Results and discussion

Here, we report the model performance of our atmospheric chemistry prediction models for  $NO_x$  and the accompanying regression model for the  $NO_2$ :NO ratio that enables us to convert  $NO_x$  columns to  $NO_2$  columns observed by satellites. We assess the fidelity of our results from these models using the full-chemistry version of GEOS-Chem and evaluate our results using TROPOMI  $NO_2$  column data.

## 3.1 Performance of atmospheric chemistry regression models for NO<sub>x</sub>

# 3.1.1 NO<sub>x</sub> chemistry random forest




Fig. 3a shows that the  $NO_x$  chemistry random forest model has an impressive performance at reproducing results from the full-chemistry version of GEOS-Chem for the four months we study in 2019. The model performance  $R^2$  values are 0.97, 0.97, 0.96 and, 0.95 for January, April, July, and October 2019, respectively. The MAE values are largest in July ( $4\times10^4$  molec/cm<sup>3</sup>/s) and smallest in January ( $2.3\times10^4$  molec/cm<sup>3</sup>/s), reflecting the increase in magnitude of chemistry rates during summer months over Europe.

We also tested our regression-based atmospheric chemistry model with model data from 2021 (Fig. A6). As expected, the regression model performance has less skill in reproducing data that has not been used for training. In this case, the MAE values are higher by a factor of 1.3-1.8 compared with the overall performance comparison shown in Fig. 4). Nevertheless, the model still shows substantial skill despite substantial differences in anthropogenic emissions between 2019 and 2021 due to COVID-19. Specifically,  $NO_x$  emissions were found to decrease by 18-24% during lockdown periods (Miyazaki et al., 2021) leading to a mean observed reduction in  $NO_2$  of 29% (Cooper et al., 2022).

# 3.1.2 NO<sub>x</sub> chemistry prediction using constant lifetime scaling

Fig. 3b shows results from using our alternative atmospheric chemistry regression  $NO_x$  model that employs a constant atmospheric lifetime scaling approach (eq. 4). The resulting model performance is a significant improvement above the other regression model for all four study months. Using our scaling approach, we found consistent values of  $R^2 = 1.0$  and MAE values that are approximately 2-3 times smaller than the other regression model. As with the other regression model, the size of the error is scaled by the seasonal changes in chemistry rates.

While this approach shows extremely encouraging abilities to determine  $NO_x$  chemistry rates, its effectiveness relies on having a full-chemistry model run available for at least one set of emission inputs. Consequently, this approach is particularly useful for emission perturbation studies, for which numerous emission distribution scenarios might be needed for model inversion work. In this case, the full-chemistry model would only need to be run once for the given time period of interest. However, we cannot predict the  $NO_x$  chemistry using this method for a previously unmodelled meterological period.

Figure 3. Actual versus predicted scatter plots for models tested on simulations with unseen emission perturbations. (a) The random forest regression model for predicting the  $NO_x$  chemistry rate, (b) the constant lifetime scaling for reconstructing the  $NO_x$  chemistry rate using an unperturbed chemistry dataset, (c) the reconstruction of  $NO_2$  from  $NO_x$  using the random forest regression model for predicting the  $NO_2$ :NO ratio.

# 3.1.3 NO2:NO ratio regression model







We find the random forest regression model to predict  $NO_2$ : NO ratios also demonstrates significant performance. The predicted ratio is used to convert  $NO_x$  concentrations to  $NO_2$  concentrations (eq. 2). Figure 3c shows that the regression model can reproduce "true"  $NO_2$  values from the full-chemistry of the GEOS-Chem model, with values of  $R^2$  of 1.0; the exception is January when  $R^2 = 0.99$ .

Generally, the model performance is better during summer months and worse in winter months, with MAE values an order of magnitude smaller in July compared to January. This is partly due to NO<sub>2</sub> concentrations increasing during colder months due to increased combustion and longer nights, and because we find that NO<sub>2</sub>:NO ratios become increasingly hard to determine at higher solar zenith angles, typically experienced over Europe during daytime through winter months. We also examine the performance of this regression model using data from the unseen year 2021. As with the atmospheric chemistry regression model, described above, the performance was good but worse than for 2019 in which data was used to train the model. The MAE increased by a factor of 3.25, 3.52, 3.04, and 3.14 for January, April, July, and October respectively. We found the R<sup>2</sup> performance reduced most for January from 0.99 to 0.92, During April and October R<sup>2</sup> reduced from 1.0 to 0.99, while R<sup>2</sup>=1.0 was maintained in July.

## 3.2 NO<sub>x</sub> atmospheric modelling

Fig. 5 shows the  $NO_x$  column reconstruction for the two regression models used to describe the  $NO_x$  chemistry rates from the full-chemistry version of the GEOS-Chem model. From a visual inspection, there are no obvious differences in the spatial distribution of the  $NO_x$  columns reconstructed using both the regression-based chemistry model and the constant lifetime scaling model. However, when mapping the differences, there are areas of deviation from the full-chemistry model. Broadly, this deviation is significantly smaller when we use the scaling-based model compared to the regression-based. In addition, the error accumulation in January is notably smaller than in other months.

Fig. 6 shows the temporal variation in the reconstruction error. The range, IQR, and median values are shown in 6a and the mean absolute percentage error (MAPE) is shown in 6b. For the regression-based chemistry method the range in deviation peaks at up to  $3\times10^{14}$  molec/cm<sup>2</sup> in January,  $5\times10^{14}$  molec/cm<sup>2</sup> in April and  $6\times10^{14}$  molec/cm<sup>2</sup> in July and October. This is reflected in maximum MAPE values of 2.8%, 9.7%, 8.9%, and 9.3% for the four months, respectively. On the whole, the MAPE reduces through time, with final deviation values of 1.7%, 3.4%, 2.0%, and 4.8% after the full 10-day run.

Reconstruction errors for the constant lifetime scaling model show much smaller errors, particularly in January, with MAPE < 0.2% throughout the 10-day run. This is driven by the smaller impact that emission perturbations have on the  $NO_x$  chemistry in January as shown by Fig. A5. In particular, the lifetime of  $NO_x$  is relatively unchanged between the unperturbed and perturbed model runs. This reduced impact in January is likely due to the slower rate of photochemical reactions in the winter months and increased atmospheric stability at lower temperatures. The other months do see a more prominent deviation of up to a maximum of  $4 \times 10^{14}$  molec/cm<sup>2</sup>, with peak MAPE values of 6.6%, 5.7%, and 4.5%, for April, July, and October, respectively. As with the regression-based model outputs, here the MAPE also generally decreases through time with final

**Figure 4.** Regression model prediction performance compared when tested on a 20% perturbed model run for 2019 and an unseen year, 2021. (a) Shows the NO<sub>2</sub> chemistry regression model performance comparisons and (b) shows the NO<sub>2</sub> prediction performance using the NO<sub>2</sub>:NO regression model.

deviation values of 0.1%, 1.1%, 0.2%, and 0.3% for each month, respectively. Interestingly, while the range and IQR are relatively stable throughout the run when using the regression-based reconstruction, these quantities decrease considerably with time when we use the scaling-based reconstruction.

The reconstruction error has a small diurnal cycle, peaking in the morning and to a lesser extent in the evening, reflecting the diurnal cycle of  $NO_x$  chemistry (Fig A1). Overall the absolute model error for both the regression-based and scaling-based methods peaks after the first day and then gradually reduce, plateauing by  $\simeq$ day 6. This early peak in error followed by a reduction and eventual plateau is likely due to compensating errors, where the regression model's over- and under-predictions balance each other out over time, leading to a stabilisation of the overall error. It is encouraging that there is no accumulation of error through time, suggesting this approach would be suitable for studies longer than for ten days. It is clear that the optimal reconstruction performance is found when using the scaling-based method, but as we already note there are limitations to this method. The regression-based approach still provides excellent reconstruction performance for our purposes.



To evaluate the performance of the regression-based chemistry modelling approach with regression models trained on a different meteorological time period, the same models were applied to simulate atmospheric  $NO_x$  over Europe for 2021.

Figure 5. The modelled  $NO_x$  columns sampled at 12:00 UTC after a 10-day model run with  $\pm 20\%$  emission perturbations.  $NO_x$  columns are compared for the GEOS-Chem full-chemistry model and (a)  $NO_x$  columns are simulated using the regression-based chemistry method and (b) using the constant lifetime scaling method.




Figure 7a shows the reconstructed  $NO_x$  columns after a 10 day model run. As expected, the reconstruction performance is clearly worse than when the regression-based chemistry is just applied in 2019 with emission perturbations (Fig 5a). However, from a visual inspection, there are no obvious changes to the spatial distribution of the  $NO_x$  columns reconstructed using regression-based chemistry in comparison to the full-chemistry model output. Additionally, the temporal variation in error is shown through plots of the MAPE (Fig 7b). We see maximum MAPE values of 11.0%, 10.0%, 16.7%, and for January, April, July, and October 2021 respectively. For all months this is an increase in the maximum deviation observed when applying this methodology to a perturbed 2019 run. Overall, this is reflective of the reduction in prediction power of the regression models when we apply to 2021, which has unseen meteorology. Overall, the same pattern of the absolute error gradually reducing and plateauing by  $\simeq$  day 6 is also observed here. However, the diurnal cycle of variation in the reconstruction error is more pronounced in the 2021 case, likely due to the fact that the regression model is worse performing during the night for unseen meteorology. The error tends to reduce dramatically towards the middle of the day, which is helpful if we consider the application of model comparison with satellite data such as a TROPOMI, which has a 13.30 overpass time.

Substantial computational time is saved when we employ these regression methods to model atmospheric NO<sub>x</sub>. Figure 6c shows the time taken for each model to perform a 1-day model run. This was calculated as the mean average for the model to run for a single day out of the 10 days run for each of the four months, repeated for 3 model runs. Clearly, the full-chemistry

Figure 6. Comparison of the temporal variation in  $NO_x$  column reconstruction for the regression-based and scaling-based model. (a) The median (dashed line), IQR (light-shaded region) and range (dark-shaded region) of the  $NO_x$  column reconstruction error over the 10-day runs. (b) The mean absolute percentage error over the 10-day runs. (c) Shows the reduction in computational time when modelling atmospheric  $NO_x$  using each of our chemistry prediction methods compared to running with the full-chemistry model.

model takes the longest, with a mean of 52 minutes per day for our nested model over Europe. The regression-based chemistry model is significantly faster with a mean of 16 minutes (3.25 times improvement), while the constant lifetime scaling method is even faster, with a mean of 12 minutes (4.3 times improvement). It is important to note that the model run times reported here are subject to variability due to fluctuations in the relative loading experienced by the computer system used.

# 3.3 NO<sub>2</sub> column reconstruction


Finally, we assess the capability of our NO<sub>2</sub>:NO regression model, convolved with TROPOMI instrument averaging kernels, to reproduce observation column distributions of NO<sub>2</sub> from TROPOMI. The absolute differences in NO<sub>2</sub> columns between

Figure 7. (a) The modelled  $NO_x$  columns sampled at 12:00 UTC after a 10-day model run in 2021 using the regression models trained on 2019 compared with full-chemistry. (b) The mean absolute percentage error for the 10-day runs.

GEOS-Chem full-chemistry and the GEOS-Chem regression-based and scaling-based models are compared to the absolute difference in TROPOMI NO<sub>2</sub> and GEOS-Chem full-chemistry, as well as to the magnitude of the TROPOMI NO<sub>2</sub> column precision data. This is presented in Fig. 8a, compared for 8 days in January, April, July, and October. We apply the regression-based method to a 2019 perturbed model run, and to a 2021 model run.




We find comparable  $NO_2$  reconstruction errors for the four months we study. Earlier, with the  $NO_x$  reconstruction, we found that the error was smaller for January than the other months (Fig. 3a and 3b), however, the higher error from the January  $NO_2$ :NO regression model (Fig. 3c) offsets this advantage, ultimately bringing the overall reconstruction error for all months to a comparable level. We observe comparable magnitudes of reconstruction error when we compare our  $NO_2$  reconstructions based on the scaling-based and regression-based methods applied to the 2019 model run. However, the reconstruction error tends to be consistently larger when we apply our regression-based method to the year 2021. This is particularly notable in January and July, which can be attributed to the greatest deterioration in  $NO_x$  chemistry regression performance in July 2021, and the greatest deterioration in the  $NO_2$  prediction performance in January 2021 (see Fig 4).

When we compare the difference between GEOS-Chem and TROPOMI NO<sub>2</sub> columns, we find that the NO<sub>2</sub> reconstruction errors are much smaller and much smaller than the estimated precision values for the data. This is the case for the scaling-based approach and the regression-based approach applied to both 2019 and 2021. This provides confidence that our model reconstruction performance is robust enough for use in inversion work, even in the case of using regression models that

have been trained on unseen meteorological periods. See Appendix B for a more detailed analysis on the difference between modelled column  $NO_2$  and observed TROPOMI data.

Fig. 8b, shows that the median NO<sub>2</sub> column model reconstruction errors are 2.8% of the actual deviation from TROPOMI in the scaling-based approach, compared to 6.5% and 7.3% in the regression-based approach for 2019 and 2021, respectively. Similarly, these construction errors represent a median value of 1.3% of the TROPOMI precision value for the scaling-based approach, compared to 2.9% and 3.2% for the regression-based approach for 2019 and 2021, respectively. Across all reconstructed data points, we found that over 99.9% of the data had reconstruction errors smaller than the corresponding TROPOMI column precision for both reconstruction methods in 2019. For the regression-based method applied in 2021, this was true for over 99.7% of the data.

# 4 Concluding remarks







We have demonstrated that the  $NO_x$  chemistry rates and  $NO_2$ :NO ratio described by a leading 3-D atmospheric chemistry model can be reproduced using random forest-based regression models using  $NO_x$  concentrations, the spatial location, and meteorological variables as input parameters. The models perform successfully on perturbed testing data through all months of 2019 with  $R^2 > 0.95$  for predicting  $NO_x$  chemistry rates and  $R^2 > 0.99$  for predicting the corresponding  $NO_2$ :NO concentration ratios. We also show that these models maintain their prediction capability when tested on model outputs from an unseen year (2021) with contrasting environment conditions.

We have also demonstrated that the atmospheric lifetime of  $NO_x$  is stable against varying emissions, particularly in winter months. From this, we have demonstrated that it is also possible to predict updated  $NO_x$  chemistry rates of change as a result of emission perturbations, with knowledge of  $NO_x$  chemistry from an initial unperturbed model run. This scaling-based approach has impressive prediction performance with  $R^2$ =1.0.

We have developed two viable methodologies to model atmospheric  $NO_x$  in a more computationally efficient way than using the GEOS-Chem 3-D model. The regression-based chemistry method has the advantage of not requiring prior knowledge of the  $NO_x$  lifetimes for a baseline model run, and reduces the computational time by a factor of 3.25. The lifetime scaling-based approach reduces the model run time slightly further by a factor of 4.3, but a baseline full-chemistry model run is required. This scaling-based approach has smaller model reconstruction errors, but generally both approaches have reconstruction errors smaller than the TROPOMI precision values for over 99.9% of the reconstructed data (399,502 points).

Our study provides confidence in random forest models being used to describe  $NO_x$  chemistry to a sufficient accuracy for them to play an important role in inversion methods. Previous work has already found that  $NO_2$  can be used to help constrain ffCO<sub>2</sub> (Berezin et al., 2013; Lopez et al., 2013; Goldberg et al., 2019; Super et al., 2020), and this work develops a new methodology to more efficiently infer  $NO_2$  column enhancements from changes to  $NO_x$  emission inputs. The methodologies developed here will be used within a joint  $NO_x$ : $CO_2$  model inversion to constrain geographically resolved ffCO<sub>2</sub>. This will be explored using an ensemble Kalman filter within the GEOS-Chem model framework, as well as within the Integrated Forecasting System (IFS) using an incremental 4D-Var algorithm (Inness et al., 2013). Results from our study are particularly

**Figure 8.** (a) The absolute difference in NO<sub>2</sub> between GEOS-Chem full-chemistry and the constant lifetime scaling based model (blue); the regression-based chemistry model applied to a 2019 perturbed run (green) and applied to a 2021 run (purple); deviation from the observed NO<sub>2</sub> TROPOMI columns (red); as well as the TROPOMI NO<sub>2</sub> tropospheric column precision values (yellow). (b) The normalised NO<sub>2</sub> differences are calculated by normalising the reconstructed model deviation by the absolute deviation between GEOS-Chem and TROPOMI, as well as by the TROPOMI column precision values. For the different model reconstructions, the NO<sub>2</sub> deviation is consistently less than the corresponding TROPOMI precision value in more than 99.5% datapoints.

timely with the launch in the next few years of the Copernicus Anthropogenic Carbon Dioxide Monitoring constellation (CO2M) that include column measurements of CO<sub>2</sub> and NO<sub>2</sub>. Overall this work will support the development and employment of European CO<sub>2</sub> measurement, reporting and verification systems.

# 5 Code/Data availability

The analysis code, model output data, and random forest regression models (in .pkl format) are available upon request from the corresponding author (cschooli@ed.ac.uk).

# 6 Author contributions

CS performed the GEOS-Chem model runs and data analysis. CS, PP, AV, and NB were involved in discussions and contributed to the development of the methodology. CS and PP wrote the paper. AV and NB provided feedback and comments on the paper.

# 7 Competing interests

The authors declare that they have no conflict of interest.

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

# Appendix A

Figure A1. Diurnal cycle of  $NO_x$  chemistry for four months of the year. Median and interquartile range net rates of change at the surface of the atmosphere averaged across the European domain.

**Figure A2.** a) Feature selection results for the rate prediction models, obtained using a forward selection wrapper method. Plotted are the coefficient of determination (R<sup>2</sup>) and mean absolute error (MAE) as functions of the number of features included, for each of the four seasonal models (January, April, July, October). (b) Same as (a), but for the partitioning ratio prediction models. (c) Feature importance distributions for each of the four monthly models, showing the relative contributions of each predictor variable to the rate prediction models (using nine features) and the partitioning ratio prediction models (using eight features). (d) Change in MAE resulting from the removal of each of the 14 features in turn, demonstrating the individual impact of each feature on model performance and highlighting the importance of specific predictors for accurate rate and ratio estimates.

Figure A3. Individual relationships between the nine regression input parameters and the  $NO_x$  net rate of change. A LOWESS fit (red line) illustrates smoothed trends in the data, with  $R^2$  values reported for each fit. Among the parameters,  $NO_x$  concentration, altitude, and temperature exhibit noticeable trends with chemistry rates, while the remaining parameters show little to no clear trends individually.

Figure A4. Impact of hyperparameter changes on random forest regression model performance for predicting  $NO_x$  chemistry rates. Plots show the effect of varying the number of trees, maximum tree depth, maximum leaf nodes, and maximum features per decision on mean  $R^2$ , MAE, and prediction time (shaded regions represent performance ranges across monthly models). Increased algorithm complexity improves  $R^2$  and reduces MAE but increases prediction time. Optimal hyperparameters—40 trees, depth of 30, 300,000 leaf nodes, and 4 features per decision—achieve balanced performance with a prediction time of 6 ms.

Figure A5. The spatial distribution of the impact of  $\pm 20\%$  emission perturbations on (a) the NO<sub>x</sub> net rate of change, and (b) the atmospheric lifetime of NO<sub>x</sub>. Overall, it is clear that the impact on the atmospheric lifetime is much smaller, due to its independence from the NO<sub>x</sub> species concentration. Note that a negative lifetime of NO<sub>x</sub> arises in areas where we have a net chemical production of NO<sub>x</sub>.

Figure A6. Testing the regression models on 2021. (a) The random forest regression model for predicting the  $NO_x$  chemistry rate, (b) The reconstruction of  $NO_2$  from  $NO_x$  using the random forest regression model for predicting the  $NO_2$ :NO ratio.

## 515 Appendix B: Comparison with TROPOMI

The  $NO_2$  columns modelled by GEOS-Chem was compared directly with the TROPOMI data for assessment of agreement. Scatter plots between the two are shown in Fig. B1, where we found significant Pearson correlations (p

Figure B1. Correlation between modelled GEOS-Chem  $NO_2$  columns and observed TROPOMI  $NO_2$  for the four months of interest. The Pearson rank and mean absolute area are shown in the legend. The best-fit line (red-dashed) can be compared to the y=x line (black).

Figure B2. Comparison between GEOS-Chem and TROPOMI for 5 days in January, April, July, and October.