# Peer review of "Development of a parametrised atmospheric $NO_x$ chemistry scheme to help quantify fossil fuel $CO_2$ emission estimates"

_EGUsphere, 2024_

## Author Response (AR1)

**Reviewer 1**

This paper presents a methodology for parameterizing NOx chemistry to enable the inversion of CO2, combined with NOx, for constraining fossil fuel CO2 (ffCO2) emission estimates at reduced computational cost. The authors employ a machine learning-based random forest regression model to predict the rate of change of NOx, thereby replacing the need for full-chemistry mechanism simulations during inversion. Second model estimates the NO2:NO ratio to convert satellite-based NO2 column measurements into total NOx column density. It is trained using GEOS-Chem outputs from a 2019 model run and validated against simulations with randomly perturbed anthropogenic NOx emissions, as well as a 2021 model run. Conceptually, this work on parameterizing NOx chemistry is useful to constrain ffCO2 emissions, but requires more robust model design and validation to be published.

**General Comments:**

1. Please provide a more detailed explanation of your machine learning model configuration process to ensure that readers who are not familiar with machine learning can follow. For example, include a description of the hyperparameter tuning and the forward feature selection procedures.

**A more detailed explanation of both of these processes has been included in Section 2.2.**

2. I believe the current design of the model training and testing has a significant limitation. Validating the model on a simulation with perturbed emissions but under the same atmospheric conditions as the training data may lead to overly optimistic validation results, as it does not sufficiently test the model's generalizability to independent atmospheric states. The authors do evaluate the model on a 2021 simulation; however, I recommend adopting a more rigorous validation approach. Specifically, consider randomly selecting 75% of the full dataset across both grid cells and time steps for training, and using the remaining 25% for evaluation. This would provide a more robust assessment of the model's generalizability across spatial and temporal domains.

Thank you for the valuable suggestion. We agree that validating a model only under similar atmospheric conditions as the training data could lead to optimistic results. However, in our current setup, the primary aim is to assess the model's performance under emission perturbations, to isolate the response of NOx chemistry to emissions alone. This design choice reflects our target application, which is intended for use in inversion studies where the model is rerun for a large ensemble of emission perturbation scalings. Therefore, assessing model performance under this constrained setup provides a relevant test of its predictive capability in the intended use case.

That said, we would like to emphasise that our validation approach already incorporates several elements of rigorous generalisability testing:

- We draw from a very large GEOS-Chem dataset:  $117 \times 114 = 13,338$  horizontal grid cells, across 15 vertical levels and 24 hourly time steps for every day in a given month. This amounts to approximately  $2.2 \times 10^8$  data points per month.
- For model training, we randomly sample 10% of the unperturbed dataset across all spatial locations and time steps, which ensures a representative but compact training set.
- For testing, we sample 0.25% of the perturbed dataset (~570,000 points) across the same spatial and temporal domain. These test points are drawn from a model run with different emission inputs, meaning the model is evaluated on physically different conditions it was not trained on.

- Due to the random selection of both training and testing points across time and space, the overlap in specific spatiotemporal conditions between the training and testing sets is minimal. An expected~0.025% of the test data may overlap with the training distribution in space and time.
- Importantly, the emission conditions in the test set are entirely unseen by the model, adding an orthogonal source of variability that the model must generalise across.

Thus, while we do not fully separate the training and test sets by atmospheric state (e.g., by using entirely orthogonal temporal and spatial regions), the validation set still covers a broad range of meteorological variability due to its random sampling and includes unseen combinations of space, time, and emissions. We believe that this, along with our additional validation on a completely unseen meteorological year (2021), demonstrate that these models are robust under both unseen emissions and unseen meteorological conditions.

An additional final paragraph to Section 2.2 has been added to clearly explains this validation strategy. We hope this convinces the reviewer that the validation approach is robust.

3. For model configuration, the authors tested the nine selected features along with pressure, air density, planetary boundary layer height, and the relative mixing ratio of ozone to carbon monoxide. Among these, please explain why the relative mixing ratio of ozone to carbon monoxide is expected to help predict the rate of change of NOx.

Apologies. We now realise that the wording of this description was misleading. We have adjusted the wording to be 'the mixing ratio of ozone (O3) and the mixing ratio of carbon monoxide (CO)'. As we tested two separate parameters – ozone column mixing ratio, and CO column mixing ratio (not the relative ratio between the two).

Ozone concentration affects NOx concentrations by reaction with NO to form  $NO_2$ , shifting the  $NO/NO_2$  balance through rapid photochemical cycling. Additionally, higher ozone increases the production of hydroxyl radicals (OH), which accelerates the irreversible removal of NOx via the formation of nitric acid (HNO3). Additionally, CO affects NOx concentrations by reacting with OH radicals, reducing the OH available to oxidise  $NO_2$  into nitric acid (HNO3), thereby slowing NOx loss. As a result, higher CO can increase NOx lifetime by competing for the atmosphere's oxidative capacity.

We decided to examine these input parameters because satellite retrievals for both metrics are available, making it possible to use this data to inform our model runs. However, we found that while each of these parameters had some predictive power on its own, they were not essential for the models. Including them did not improve the predictive performance when combined with other parameters, which is why they were not included in the final models. A more detailed analysis of the feature selection and performance impact of each predictor is now included in the appendix Fig. A2.

4. I would also suggest exploring additional input parameters based on our understanding of NOx chemistry. For example, incorporating time of day could help distinguish between daytime and nighttime chemical processes. Including satellite-derived variables such as O₃ column density and HCHO column density could provide insight into VOC levels, allowing the model to better account for variations in NOx chemistry under different VOC conditions. Variables such as J(O¹D) and H₂O concentrations could help to better represent hydroxyl radical (OH) levels, which play a key role in NOx chemistry.

Thank you for the suggestions. We agree that the proposed variables are highly relevant to NOx chemistry rates and could serve as informative predictors. In our current setup, we included water vapor (H2O) as a volume mixing ratio and solar zenith angle (SZA), which served as proxies for

photochemical activity and time of day. These were selected to help capture aspects of OH variability in the absence of explicit radical chemistry.

As noted in response to Comment 3, we tested the O3 column mixing ratio was tested but we found that it did not improve model performance during feature selection and was thus excluded for parsimony. While HCHO column density is often a useful proxy for VOC reactivity, our initial tests found it to be weakly correlated with NOx chemistry rates in this setup (for example, much lower correlation compared to O3 and CO columns). Regarding J(O¹D), we agree it is an important driver of OH production. However, because our models are designed to operate on simulations with chemistry mechanisms disabled—including OH and related photolysis reactions—J(O¹D) is not available. Our overarching goal is to develop a machine learning parameterisation for NOx chemistry that relies only on variables accessible in offline model configurations, primarily meteorological and spatiotemporal features.

We hope the reviewer agrees that the selected input parameters strike a balance between physical relevance and practical availability and that the resulting model performance demonstrates the viability of this approach for approximating full-chemistry outputs in a computationally efficient way.

I also recommend providing a more detailed description of the parameter selection criteria, starting with an analysis of the impact of individual parameters on model performance. It would be helpful to show how the inclusion of each parameter incrementally improves the model's accuracy.

A more detailed description of the parameter selection criteria has been included in Section 2.2, specifically the method is a forward selection wrapper based on MAE. We have included plots showing how incrementally adding the parameters improved model performance for the chemistry rate prediction model (A2a) and the partitioning ratio prediction model (A2b). In addition, the feature importances, and the change in model performance when each feature is removed are presented in A2c and A2d, respectively. We hope this additional analysis helps the reviewer understand the feature selection process and the importance of the different predictors.

**Minor Comments:**

- Line 35: correct to methodology

**This has been corrected**

- Line 107-108: please use NO2+O2♦NO+O3

Thank you for pointing this out, the equation has been corrected.

**Reviewer 2**

The authors report on a method to derive NOx concentrations and NO2:NO ratios, from a machine-learning method which uses as input variables apart from NOx concentrations the meteorological factors and location, as well as a reaction rate scaling method.

They describe their approach, and validate this against simulations done with the parent CTM (GEOS-Chem), both for the same year but perturbed emissions, and for an alternative year (with alternative meteorology).

Overall, both methods were shown to successfully capture the simulated NOx, as well as simulated NO2 columns when compared to the full-chemistry solution. The errors in NO2 columns were reported typically an order of magnitude smaller than the differences between GEOS-Chem and TROPOMI, and the TROPOMI column precision.

I recommend publication in ACP after addressing the comments given below.

The Introduction could be strengthened, by checking and adding a few more references, as it appears that the authors make quite a few statements that are not well backed up with the given references, and in cases the formulation is a bit sloppy.

We agree that the Introduction had been lacking some references to support key statements. We have carefully reviewed the section to identify unsupported claims and included a number of additional references to support these. Additionally, we have reformulated a few sentences and reordered some paragraphs to improve the clarity and flow. We hope these revisions ensure that the motivation for our study is clearly presented.

In my view a further discussion on the selection of input parameters (Table 1) is welcome. e.g. What happens if the longitude/latitude is excluded from the parameter list? the thing is that by including this you make implicit assumptions on local conditions in the model, including local emissions, which is actually exactly what you want to derive from the model. This location dependence makes the model implicitly to account for local conditions, but also suggests that the method cannot directly be adopted for other places in the world, as I understand?

The reviewer is correct that the location dependence of our models is a limiting factor, implying that the models are influenced by local conditions, potentially capturing local emission patterns and atmospheric characteristics specific to the training domain. This dependence means that the model's direct applicability outside the trained region is limited without retraining or adaptation.

To address this, we have now included an assessment of the impact on model performance when each predictor is removed (Fig. A2d), this includes longitude and latitude. Our results show that, although removing either coordinate leads to a measurable increase in the mean absolute error (MAE), their overall importance is relatively small compared to other predictors. Specifically, for the chemistry rate model, excluding longitude results in a minor performance degradation across all months, while latitude removal has a more noticeable impact, particularly during colder months (January and October). For the partitioning ratio model, both longitude and latitude contribute minimally to the model's predictive skill.

We argue that the inclusion of location variables is justified and beneficial because local geographic coordinates implicitly capture spatial gradients in land cover, and regional meteorological patterns that affect chemical processes. For example, latitude often correlates with solar zenith angle, temperature gradients, and seasonality—factors that strongly influence photochemistry and reaction

rates. Longitude can capture east—west variations in emission sources or regional transport patterns that are otherwise difficult to parameterize explicitly.

Importantly, despite this location dependence, we demonstrate that the models maintain strong predictive performance when tested on substantially perturbed emission scenarios (as in Fig. 3 where the emission pattern with location changes from training to testing), and on an independent temporal period (as in A6, for 2021) which inherently uses unseen meteorology relative to location. This indicates that the model generalises well within the spatial domain and across varying atmospheric conditions relevant to the studied region.

For future work, expanding these models to application outside the European domain we will explore whether removal of the location dependence would be beneficial.

Also can you provide a metric that actually describes the importance of each of the physical input quantities. You briefly describe also other parameters (line 131-133) for the training, but exclude them. I'd be interested in a more detailed description / quantitative assessment to underline the arguments why the current selection of training parameters was made.

We have now included a plot showing the relative importance of each predictor in each of our ML models (Fig. A2c). Additionally, as described above in response to reviewer 1 we have added a more detailed description of how we performed feature selection in section 2.2, and shown the impact of feature selection plotting how incrementally adding the parameters improves model performance for the chemistry rate prediction model (A2a) and the partitioning ratio prediction model (A2b).

In figure 3 the authors present the skill of the trained model against the actual rates, i.e. from quick reading it appeared that the same data that was used for the training, while in Figure A5 in the Appendix they present the same results, but on more independent data (i.e. data for another year). I would argue to swap the figures, i.e. to show Figure A5 in the main body (which is the more important independent evaluation) while moving the intermediate result, which shows the minimal validity of the regression model to be functional) towards the appendix. Then this allows to report on the correlation performance using the more independent evaluation with a bit more emphasis. This then also allows to trace back the number reported in the Abstract (R2>0.79) in the main body of the text.

Re-reading, I Realize that Figure 3 was created with the perturbed emissions experiments, so there is some modification compared to the training dataset, if I understand well. Please update the figure legend to point this out, for the reader.

We have revised the caption of Fig. 3 to make it clear that this analysis is performed on unseen perturbed emission experiments. We have also included an additional paragraph at the end of section 2.2, which outlines our training / testing process. We hope that our response to reviewer 1 point 2 helps explain the motivation of using the perturbed emissions experiments as the main analysis for this paper, with inversion analysis being the primary planned application of this work.

Line 171: " so, if the concentration doubles then we assume a doubling in the net chemical rate of change". . Here the authors discuss the characteristics of the lifetime / reaction rate scaling method. I'm a bit puzzled by the use of negative lifetimes. Especially small negative lifetimes, which likely happen to occur now and again in the results, sounds like a recipe of blowing up the model, in case the rate of change is positive (i.e. a net increase in NOx). Could you please elaborate?

We understand why this may be confusing. In our framework, a negative atmospheric lifetime does not represent a physical lifetime in the conventional sense but rather an apparent or effective

lifetime derived from the instantaneous net chemical tendency. Specifically, a negative chemical lifetime just reflects an instantaneous net production of NOx rather than a loss. This occurs when the chemical production rate exceeds the loss rate at a given grid point in space and time, resulting in a positive net rate of change and therefore a negative calculated lifetime (since lifetime =[NOx]/(-d[NOx]/dt)). Using these negative lifetime values in the scaling method does not destabilise the model or cause it to "blow up." Instead, it leads to a scaled net chemical increase in NOx when the new concentration is higher than the original, consistent with what the local chemical tendency already indicates. Furthermore, these cases are typically much less frequent than cases of net loss and associated with specific chemical or meteorological conditions. The magnitude and frequency of these negative lifetimes are small enough that they do not dominate the overall behaviour (e.g. we do not see a dominant net production of NOx in our lifetime-scaled approach). We hope this reassures the reviewer that this lifetime scaling-based method does not lead to blowing up of NOx concentrations even when some negative effective lifetimes do occur.

We have added a clearer narrative within section 2.3 that what we calculate here is an effective lifetime, and therefore should not be interpreted as the intrinsic first-order decay timescale for NOx.

**Smaller comments:**

Lines 21 - 29: Many statements made by the authors, but I miss proper referencing here, e.g. wrt the availability of bottom-up inventories.

**References added**

line 21 "reaching net zero": net zero what? please use more explicit description.

**Clarified 'net zero greenhouse gas emissions'**

line 23 "how does a country know" this is also a bit sloppy formulation of the question, to my taste. please reformulate

**Reworded: 'how can a country assess whether'**

line 25 uncertainty -> "uncertain"

**Changed**

line 29: "subject to uncertainties": can you add references here of key uncertainties?

**Added**

line 33: you jump directly to your method, without proper introduction of the link between NOx and CO2. I would expect such a more expicit introduction, + references to past studies who have made attempts in this direction.

We agree with this point that there is a feeling of jumping straight to the method before an explicit introduction of the link between NOx and CO2. In the first paragraph we think it is useful to introduce an overview of the main underlying aims of this paper (developing computationally efficient models of NOx chemistry), but we have adjusted wording prior to this so it should be clearer to the reader why we use NOx and how it relates to CO2 emissions. We have then added more detail and references to the second paragraph of the introduction which describes the use of NOx as a proxy and discusses

previous NOx:CO2 work. The third paragraph then goes on to introduce our specific methodology in more detail.

line 35 "methodlogy"

**Fixed**

line 41, please check sentence.

**Restructured for clarity.**

line 45 "to facilitate the production of NO (..)", suggest to add the phrase something like ".. which is therefore co-emitted with the CO2 emissions".

**Added**

line 47 "parent emissions" please change to "parent NOx emissions". the linking to CO2 is yet another step that deserves its own discussion, to my taste

**Changed**

line 50 "becomes a widely used approach", but please also give an overview of the main issues, apart from the computational costs. This is missing so far.

We have included a couple of sentences on the main issues and sources of error for these methods, largely driven by the uncertainty in NOx:CO2 emission ratios.

Figure 1 - this figure is introduced at the end of the introduction - but the steps are difficult to follow. Please expand the description of this figure on line 64, or refer more specifically to the various steps in this figure in the consecutive subsections.

We have expanded on the explanation of this figure where it is introduced.

line 75 "or a scaling-based method" is this method described further down the text? If not, remove reference to this..

This is described below (see section 2.3), but we have now explicitly introduced the name 'constant lifetime scaling-based' to describe this method.

line 78 "for data assimilation": add "on a high horizontal resolution"?

**We have re-worded as high resolution data assimilation**

line 81: add a reference to a default description of GEOS-Chem (paper? Website? other?)

**Added**

line 84: "30 models below tropopause" given the importance of near-surface processes, can you specify the depth of the first model layer? I think this is larger than the one used operationally in IFS in its default vertical resolution (10 m, see, .e.g. https://confluence.ecmwf.int/display/UDOC/L137

+model+level+definitions) - would that have implications on the accuracy of simulating NOx chemistry, surface fluxes, and dynamics?

The depth of the first model layer for GEOS-Chem is 130-180m depending on the local surface pressure, as the reviewer states this is larger than that for IFS. The coarse first layer in GEOS-Chem will reduce the ability to resolve steep gradients near the surface, potentially smoothing NOx concentrations and affecting flux coupling. However, the impact on our results is expected to be minimal because we compare model output with satellite observations of NO2 total tropospheric columns (TROPOMI), which are primarily sensitive to integrated vertical amounts rather than fine-scale near-surface gradients. Model comparison with other atmospheric models such as IFS is certainly of interest, and we do have ongoing work comparing NOx chemistry parametrisation between the GEOS-Chem and IFS models.

line 110: can you discuss, and explain the shape of the diurnal cycle in the NOx tendency over the season? e.g. what explains the large sink in NOx during night-time?

A description of this has been added.

Figure A3: please check the x-axis for the temperature plot.

Thank you, there was an error in what we were plotting here and this has been fixed.

Line 156 "the atmospheric lifetime becomes negative": it is unclear to me what is the physical meaning of a negative lifetime, as well as any possible implications on the model results. Could you please elaborate, especially in the case of small negative lifetimes?

The above response to question regarding line 171 should hopefully help answer this for the reviewer. Overall, it is important to understand that the effective lifetime we calculate is derived from the instantaneous net chemical rate of change and therefore does not correspond to an intrinsic first-order decay timescale for NOx. Instead, it reflects the net tendency at that moment, incorporating the combined effects of production and loss pathways under the prevailing chemical and meteorological conditions. As such, while useful for scaling purposes, it should not be interpreted as a purely physical or mechanistic lifetime. We have added a couple of sentences in section 2.3 to make this clear.

Figure 2b: as many the points are plotted on top of one another, it might be better to present this information in terms of a scatter density plot. Also, the judgement when delta-NOx changes are considered irrelevant appears a bit 'ad hoc'

We thank the reviewer for their suggestion of this, we have remade these figures as scatter density plots, which definitely show the overall trends more clearly. We have also added the definition of irrelevant or negligible delta-NOx changes, which we defined as being the point where the percentage change to chemistry rate was within 1%. Specifically, this looks like the absolute change in chemistry rates being <9E3 molec/cm3/s.

line 182 "below" => Above ?

Changed

**List of relevant changes**

**Introduction**

- More references have been included to support statements.
- The general background on using NOx to constrain ffCO2 has been expanded, with additional references and a description of its limitations.
- A step-by-step methodology presented in Figure 1 is now described in greater detail.

**Data and methods**

- Included the depth of the first model layer in the description of the GEOS-Chem setup.
- Corrected an error in the equation describing the photochemical decay of NO2.
- Added a description of the diurnal cycle of NOx (with reference to Fig. A1).
- Expanded Table 1 to include all 14 parameters tested for model development, along with the outcomes of feature selection indicating which parameters were used for each model.
- Changed Fig. 2b to density scatter plots and expanded the caption to explain the definition used for negligible changes to chemistry rates (<1%).
- Added more details on the feature selection procedure and hyperparameter tuning (with reference to additional Fig. A2).
- Added more details on the training and testing setup used for validation with unseen emissions and unseen meteorology.
- Expanded the description of the NOx effective lifetime, clarifying how it differs from a physical decay timescale for NOx.

**Results and Discussion**

- An introductory sentence has been added to the caption of Fig. 3 to make it clear that it shows model tests on unseen emission perturbation outputs.

**Appendix**

- Fig. A2 has been added, showing: a) plots demonstrating how feature selection optimised model performance; b) the distribution of feature importance for each of the seasonal models under both prediction schemes; and c) the effect on model performance of removing each parameter in turn from the models.
- The temperature plot in Fig. A3 has been corrected.

---

## Author Response (AR2)

**Dear Beatriz Monge-Sanz,**

Thank you for your comments. We have now added a paragraph in Section 2.2 of the manuscript (Lines 142-151) providing a brief description of random forests, including how they work, the concept of a forest of decision trees, and their suitability for predicting  $NO_x$  chemistry rates and  $NO_2/NO_x$  ratios. We hope this addition clarifies the method for readers less familiar with machine learning.

We are grateful for your time and consideration and look forward to our work being published in ACP.

Kind regards, Chlöe Schooling (on behalf of all co-authors)